# 16.8% Monolithic all-perovskite triple-junction solar cells via a universal two-step solution process

Junke Wang 📧 [1], Valerio Zardetto[2], Kunal Datta[1], Dong Zhang[1,2], Martijn M. Wienk[1] & René A. J. Janssen 📧 [1,3 ✉]

Perovskite semiconductors hold a unique promise in developing multijunction solar cells with high-efficiency and low-cost. Besides design constraints to reduce optical and electrical losses, integrating several very different perovskite absorber layers in a multijunction cell imposes a great processing challenge. Here, we report a versatile two-step solution process for high-quality 1.73 eV wide-, 1.57 eV mid-, and 1.23 eV narrow-bandgap perovskite films. Based on the development of robust and low-resistivity interconnecting layers, we achieve power conversion efficiencies of above 19% for monolithic all-perovskite tandem solar cells with limited loss of potential energy and fill factor. In a combination of 1.73 eV, 1.57 eV, and 1.23 eV perovskite sub-cells, we further demonstrate a power conversion efficiency of 16.8% for monolithic all-perovskite triple-junction solar cells.

[1] Molecular Materials and Nanosystems, Institute for Complex Molecular Systems, Eindhoven University of Technology, partner of Solliance, P.O. Box 513, 5600 MB Eindhoven, The Netherlands. [2] TNO, partner of Solliance, High Tech Campus 21, Eindhoven 5656 AE, The Netherlands. [3] Dutch Institute for Fundamental Energy Research, De Zaale 20, 5612 AJ Eindhoven, The Netherlands. ✉email: r.a.j.janssen@tue.nl

Over the last decade, hybrid perovskites have been under the spotlight of the photovoltaic (PV) research community for their excellent optoelectronic characteristics, cost-effectiveness as well as solution processability[1–3]. The record power conversion efficiency (PCE) of single-junction perovskite solar cells (PSCs) has now increased to 25.2%, approaching the state-of-the-art inorganic PV cells of 29.1% and the Shockley-Queisser (S-Q) efficiency limit of ~33%[4–7]. Further increase of efficiency of PSCs alongside high-throughput and low-cost manufacturing processes provides enormous potential for commercializing perovskite PV technologies[8].

Fundamentally, the PCE of single-junction solar cells is limited by the thermalization loss of photons with energy higher than the bandgap and the transmission loss of photons with energy lower than the bandgap[8,9]. By strategically stacking two or more light-absorbing layers with complementary bandgaps, monolithic multijunction solar cells can effectively mitigate these losses and raise the theoretical efficiency limit to 68%[10,11]. In practice, III–V crystalline semiconductors have demonstrated high PCEs of 39.2% and 37.9% in a six- and triple-junction solar cell, respectively, however their intricate and costly deposition processes prohibit large-scale applications[4,12,13]. Alternative technologies such as inexpensive organic semiconductors have also been exploited for multijunction solar cells[9]. Nevertheless, given the lack of comparably high-performing organic absorber layers over a wide range of bandgaps, suboptimal PCEs of 17.4% for tandem[14], 11.6% for a triple cell[15], and 7.6% for a quadruple-junction cell[16] have been reported in such multijunction approach. Perovskite semiconductor, by virtue of its cost-effectiveness and widely tunable bandgaps[17], holds a unique promise for the development of all-perovskite multijunction solar cells. The bandgap of Pb-based perovskite can be continuously tuned from 1.5 eV to 2.3 eV by substituting I with Br[18,19], and bandgaps as narrow as 1.2 eV are obtained when mixing Pb- with Sn-based compounds[20,21]. Device and optical modeling[10,22,23] have suggested that a monolithic tandem with 1.8 eV wide-bandgap and 1.2 eV narrow-bandgap perovskite materials can reach a feasible PCE of 33.4%. Moreover, the monolithic all-perovskite triple-junction solar cell comprising 2.0 eV, 1.5 eV, and 1.2 eV absorbers leads to an even higher PCE of 36.6%[22]. To date, tremendous research effort has been made for all-perovskite tandem devices[23–29], with PCEs up to 24.8% achieved by Tan and co-workers[30]. In comparison, all-perovskite triple-junction solar cells remain largely unexplored, with only a proof-of-concept 6.7% triple cell demonstrated by Snaith and co-workers[31].

It is not trivial to fabricate monolithic all-perovskite multijunction solar cells, bearing in mind the optical and electrical losses inherent to the complex cell design and the processing and compatibility issues encountered in depositing widely different materials on top of each other[8]. Between series-connected sub-cells, the interconnecting layers (ICLs) should serve as a physical barrier to protect the underlying layer from solvents used for the subsequent layers[26]. Also, reflective losses and parasitic absorption of the ICLs need to be minimized such that more low-energy photons can reach the next narrower-bandgap absorbing layers[10]. Besides, the ICLs should possess a large sheet resistance while retaining sufficient mobility such that selected holes and electrons from the adjacent perovskite layers can recombine efficiently[26,27]. For all-perovskite multijunction cells, examples of solution-processed ICLs are p-doped cross-linked poly(triarylamine) (PTAA)/n-doped phenyl-$C_{61}$-butyric acid methyl ester (PCBM)[32] and poly(3,4-ethylene dioxythiophene):polystyrene sulfonate (PEDOT:PSS)/indium tin oxide (ITO) nanoparticles[31]. In comparison, using sputtered ITO and indium zinc oxide (IZO) as the recombination layers has enabled higher efficiency in tandem devices[25–28], albeit the increased lateral shunt pathways and

optical losses in the near-infrared region[27]. Moreover, a thin metal oxide layer such as $SnO_2$ and Al-doped ZnO (AZO) prepared by atomic layer deposition (ALD) was necessary to prevent sputter damage[24–26,28]. It has been previously demonstrated that tuning the growth conditions can yield compact and conductive ALD layers, which alone prevent solvent damage and allow for fast charge transport after depositing a thin TCO layer[26,33,34]. Recently, Tan and co-workers[30] utilized TCO-free ICLs based on $C_{60}$/ALD-$SnO_2$/Au/PEDOT:PSS for all-perovskite tandem solar cells. Nevertheless, conventional ALD technique requires vacuum and is limited by a low deposition rate. Atmospheric pressure spatial-ALD (SALD) can be done at a much higher deposition rate while preserving conformal and pinhole-free depositions, which is closer to the industrial manufacturing requirements[35–38]. Furthermore, to maximize the performance of multijunction solar cells, stringent bandgap and thickness optimizations are needed to balance light absorption and match current density among sub-cells[10,12]. However, integrating several very different perovskite layers in such a complex multijunction cell could impose significant processing challenges, as they typically require very specific film formation strategies to achieve high efficiencies in single-junction solar cells[39]. To this end, using a simple and yet effective fabrication method suitable for various perovskite compositions and bandgaps would greatly benefit the development of all-perovskite multijunction solar cells.

Here we present a versatile two-step solution process for high-quality perovskite thin films. With only minor changes in the processing conditions, we fabricate efficient single-junction PSCs with bandgaps of 1.73 eV, 1.57 eV, and 1.23 eV. Through optimization of the ICLs based on fullerene/spatial ALD (SALD) grown $SnO_2$/PEDOT:PSS, we achieve PCEs of above 19% for monolithic all-perovskite tandem solar cells consisting of 1.73 eV and 1.23 eV absorber layers. Following the same strategy, we demonstrate efficient and reproducible all-perovskite triple-junction solar cells combining 1.73 eV, 1.57 eV, and 1.23 eV absorber layers. The best-performing triple-junction device shows a very promising PCE of 16.8%, with a short-circuit current density ($J_{sc}$) of 7.4 mA cm$^{-2}$, an open-circuit voltage ($V_{oc}$) of 2.78 V, and a fill factor (FF) of 0.81.

## Results

**Formation of wide/mid/narrow bandgap perovskite films**. We focused on a mixed perovskite composition of $Cs_z(FA_{0.66}MA_{0.34})_{1-z}$ $Pb_{1-x}Sn_xI_{3-y(1-z)}Br_{y(1-z)}$ (FA = formamidinium, MA = methylammonium). By changing the molar ratio of precursor solutions, perovskites based on wide bandgap $Cs_{0.1}(FA_{0.66}MA_{0.34})_{0.9}PbI_2Br$, medium (mid) bandgap $FA_{0.66}MA_{0.34}PbI_{2.85}Br_{0.15}$, and narrow bandgap $FA_{0.66}MA_{0.34}Pb_{0.5}Sn_{0.5}I_3$ are obtained. In a two-step deposition route, inorganic salts (CsI, $PbI_2$, and $SnI_2$) dissolved in N,N-dimethylformamide (DMF) and dimethyl sulfoxide (DMSO) are first spin-coated to obtain an intermediate precursor film, on which organic salts (FAI, FABr, MAI, and MABr) dissolved in isopropanol are spin-coated and followed by thermal annealing to accelerate the transition to perovskite crystals (Supplementary Fig. 1). Similar processing conditions were used for all the three perovskite recipes, except for a room temperature drying process of Sn-containing precursor film before the second deposition step[40]. UV-vis-NIR absorption and photoluminescence (PL) spectra indicate bandgaps of 1.73 eV for $Cs_{0.1}(FA_{0.66}MA_{0.34})_{0.9}PbI_2Br$, 1.57 eV for $FA_{0.66}MA_{0.34}PbI_{2.85}Br_{0.15}$, and 1.23 eV for $FA_{0.66}MA_{0.34}Pb_{0.5}Sn_{0.5}I_3$ perovskite absorbers, respectively (Fig. 1a). Also, the 1.73-eV perovskite film exhibits a good photo-stability by retaining its PL profile after 120 min of continuous illumination (Supplementary Fig. 2). X-ray diffraction (XRD) patterns confirm the formation of single-phase crystallites among all perovskite films (Fig. 1b).

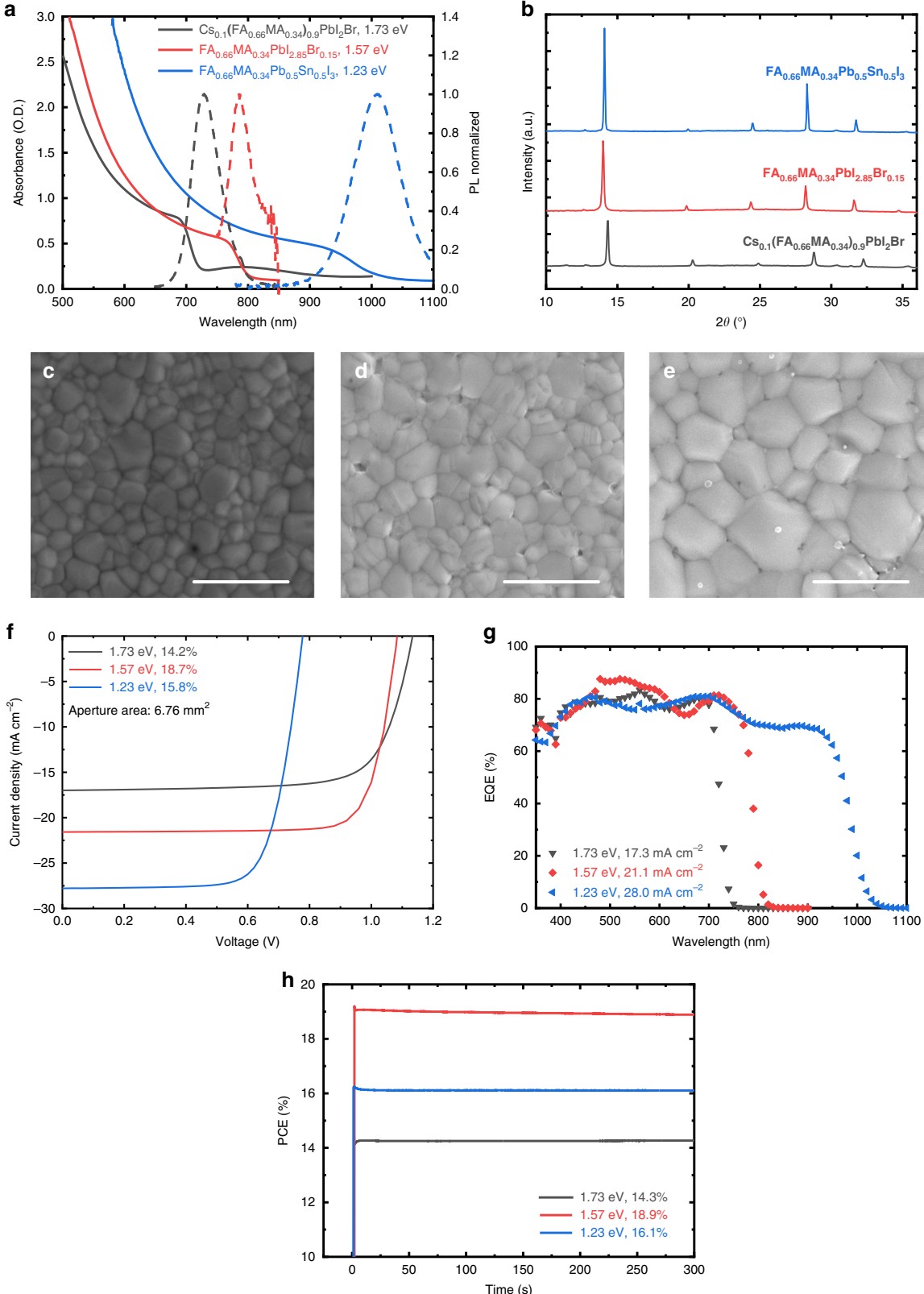

**Fig. 1 Film characteristics and device performance of 1.73 eV, 1.57 eV, and 1.23 eV perovskites prepared by a two-step solution method. a** UV-vis-NIR absorption spectra and steady-state photoluminescence. **b** XRD patterns of perovskite films with different bandgaps. **c**–**e** Top-view SEM images of 1.73 eV, 1.57 eV, and 1.23 eV perovskite films. Scale bars are 1 μm. **f** Stabilized J–V curves (measured with 6.76 mm$^2$ aperture area). **g** EQE spectra. **h** Steady-state power output tracking for opaque PSCs with different bandgaps. Source data are provided as a Source Data file.

Compared to 1.57 eV $FA_{0.66}MA_{0.34}PbI_{2.85}Br_{0.15}$, the shift of the (100) diffraction peak towards higher angle is consistent with a decrease in the cubic lattice constant from 6.324 Å to 6.303 Å for narrow bandgap $FA_{0.66}MA_{0.34}Pb_{0.5}Sn_{0.5}I_3$ (Sn incorporation), and to 6.194 Å for wide bandgap $Cs_{0.1}(FA_{0.66}MA_{0.34})_{0.9}PbI_2Br$ (Cs and Br incorporation; Supplementary Fig. 3 and Supplementary Table 1). It is also found that the (100) peak intensity of the narrow bandgap layer is significantly higher than the other films, in line with the fast-crystallizing property of Sn-based perovskites[40]. Top-view scanning electron microscopy (SEM) images reveal a compact and pinhole free surface morphology for all perovskite films, with an average grain size span from ~250 nm for both wide and mid bandgap absorbers, to ~500 nm for narrow bandgap perovskite (Fig. 1c–e and Supplementary Fig. 4). The film characteristics suggest the controlled formation of high-quality perovskite absorbers with different bandgaps using a two-step solution process.

We fabricated planar p-i-n PSCs to evaluate the PV performance of different perovskite absorbers. Here, both the wide and mid bandgap PSCs used a device configuration of ITO/PTAA/perovskite/PCBM/LiF/Al, whereas an ITO/PEDOT:PSS/perovskite/$C_{60}$/bathocuproine (BCP)/Ag design was used for narrow bandgap PSCs. All perovskite layers are 400–450 nm thick. Figure 1f, g display the stabilized current density–voltage (J–V) curves and external quantum efficiency (EQE) spectra of representative PSCs with various perovskite bandgaps. The corresponding PV parameters are summarized in Table 1. The 1.73-eV wide bandgap PSC shows a PCE of 14.5%, with a $J_{sc}$ of 17.3 mA cm$^{-2}$, a $V_{oc}$ of 1.13 V, and an FF of 0.74. In comparison, the device based on 1.57 eV mid bandgap exhibits an increased $J_{sc}$ of 21.1 mA cm$^{-2}$, a decreased $V_{oc}$ of 1.08 V, and an FF of 0.80, resulting in a PCE of 18.3%. The changes in $J_{sc}$ and $V_{oc}$ are attributed to the decreased perovskite bandgap, in line with the redshifted EQE onset from 720 nm to 810 nm. As expected, further decreasing the bandgap to 1.23 eV leads to a higher $J_{sc}$ of 28.0 mA cm$^{-2}$, a lower $V_{oc}$ of 0.78 V, and an FF of 0.73, yielding a PCE of 15.9% for the narrow bandgap PSC. Meanwhile, all the devices show low hysteresis between reverse, forward, and stabilized J–V scans (Supplementary Fig. 5). The PCEs of J–V measurements are further confirmed by steady-state power output tracking at the maximum power point, where all the devices show a negligible decrease in performance during the tracking period (Fig. 1h).

**ICL optimization for all-perovskite tandem solar cells.** To enable the current matching condition in multijunction solar cells as outlined above, we prepared semi-transparent PSCs with reduced thicknesses of 1.73 eV perovskite absorbers using the two-step process (Supplementary Fig. 6). As expected, the EQE-integrated $J_{sc}$ shows a decrease from 14.7 mA cm$^{-2}$ for a ~400-nm-thick perovskite film, to 13.6 and 8.9 mA cm$^{-2}$ for 300 and 100-nm-thick absorber layers, respectively (Supplementary Table 2). Here, the drop in EQE mainly occurs in the wavelength range of 500–720 nm due to reduced light absorption (Supplementary Fig. 7). In comparison, the changes in $V_{oc}$ and FF are relatively small for different

layer thicknesses, which results from similar film quality, as evidenced by SEM and XRD measurements (Supplementary Fig. 8).

We proceeded to construct all-perovskite tandem solar cells using a 300-nm 1.73 eV wide bandgap front cell and a 450-nm 1.23 eV narrow bandgap back cell. In the preliminary test, a combination of PCBM (80 nm)/SALD-$SnO_2$ (45 nm)/PEDOT:PSS layers were used as the ICLs. Similar to previous reports[26,33], we found that the SALD-$SnO_2$ deposited on a fullerene layer can significantly reduce $H_2O$ and DMF permeation, which prevents damage to the wide bandgap perovskite film caused by the solution processing of PEDOT:PSS as well as narrow bandgap perovskite layers (Supplementary Fig. 9). Also, the addition of such a relatively thin SALD-$SnO_2$ layer results in a comparably efficient single-junction 1.73 eV PSC (Supplementary Fig. 10). In the tandem cell, the EQE integrated $J_{sc}$ of 1.73 eV front sub-cell and 1.23 eV back sub-cell are 14.4 and 14.0 mA cm$^{-2}$, respectively, indicating a reasonably well-matched current density between the two sub-cells (Fig. 2a). The corresponding J–V curves show a PCE of 16.8% under reverse scan, with a $J_{sc}$ of 14.0 mA cm$^{-2}$, a $V_{oc}$ of 1.81 V, and an FF of 0.66 (Fig. 2b and Supplementary Table 3). Here, the relatively low FF and $V_{oc}$ are caused by an s-kink near the open-circuit voltage, which suggests the formation of electronic barriers in the ICLs[27,33].

We first replaced the solution-processed PCBM (80 nm) by a thermally evaporated $C_{60}$ layer (20 nm), which shows comparable PCE in single-junction PSCs (Supplementary Fig. 11). Compared to PCBM, a thin and yet compact $C_{60}$ layer has higher electron mobility[41], which can reduce charge accumulation in the ICLs. As a result, the tandem cell based on $C_{60}$/SALD-$SnO_2$/PEDOT:PSS produces a much higher FF of 0.77, an improved $V_{oc}$ of 1.91 V, a $J_{sc}$ of 13.1 mA cm$^{-2}$, achieving a PCE of 19.3% under reverse scan (Fig. 2b). In this case, the decrease in $J_{sc}$ is due to a current-limiting 1.23 eV back sub-cell (13.1 mA cm$^{-2}$), which is caused by the change in optical interference after replacing the PCBM by $C_{60}$ (Fig. 2c and Supplementary Fig. 12). On the other hand, it has been reported that a low carrier density ALD-$SnO_2$ layer may form a non-ohmic contact at the interface[33]. In comparison, we do not observe a severe s-kink in the J–V curve, suggesting that our ICLs are more conductive and thus provide decent FF also without additional layer[33]. Nevertheless, we found that the device performance can be further improved after inserting a thin Au layer (~1 nm) at the ALD-$SnO_2$/PEDOT:PSS interface, similar to a previous study[30]. As shown in Fig. 2b, the tandem device comprising $C_{60}$/SALD-$SnO_2$/Au/PEDOT:PSS ICLs exhibits an improved PCE of 19.7% under reverse scan, thanks to a higher FF of 0.82, together with a $J_{sc}$ of 12.7 mA cm$^{-2}$, and a $V_{oc}$ of 1.91 V. Notably, the extra Au layer reduces transmission in the near-infrared, which further reduces the $J_{sc}$ in the 1.23 eV back sub-cell (Fig. 2d).

Figure 3a, b display the device configuration and cross-sectional SEM of the optimized tandem solar cell. In the stabilized J–V measurement, the tandem shows a PCE of 19.2% (6.76 mm$^2$ aperture area), with negligible hysteresis between reverse and forward scans (Fig. 3c, d and Table 2). A stabilized PCE of 19.5%

**Table 1 Photovoltaic parameters of representative single-junction PSCs with different bandgaps.**

| Bandgap (eV) | $J_{sc}$ (mA cm$^{-2}$) | $V_{oc}$ (V) | FF | PCE[a] (%) | $J_{sc}$[b] (mA cm$^{-2}$) | PCE[c] (%) |
|---|---|---|---|---|---|---|
| 1.73 | 17.0 | 1.13 | 0.74 | 14.2 | 17.3 | 14.5 |
| 1.57 | 21.6 | 1.08 | 0.80 | 18.7 | 21.1 | 18.3 |
| 1.23 | 27.8 | 0.78 | 0.73 | 15.8 | 28.0 | 15.9 |

[a]The data was extracted from stabilized J–V curves under simulated AM 1.5G illumination (100 mW cm$^{-2}$). The aperture area was 6.76 mm$^2$.
[b]Calculated by integrating the EQE spectrum with the AM1.5G spectrum.
[c]Corrected PCE obtained by calculating the $J_{sc}$ integrated from EQE spectrum and $V_{oc}$ and FF from the stabilized J–V measurement.

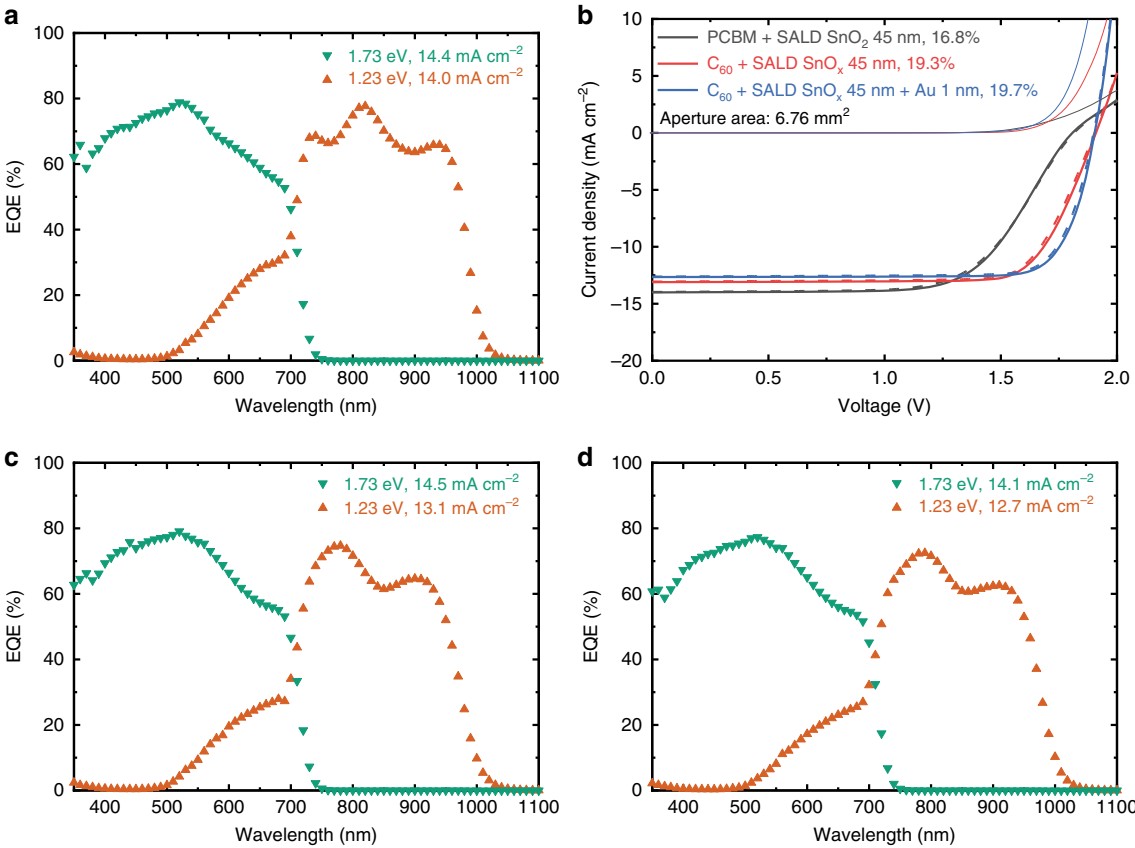

**Fig. 2 Photovoltaic performance of monolithic all-perovskite tandem solar cells.** EQE spectra of 1.73 eV wide bandgap and 1.23 eV narrow bandgap sub-cells in a tandem device. **a** For a PCBM/SALD-SnO$_2$/PEDOT:PSS, ICL. **c** For a C$_{60}$/SALD-SnO$_2$/PEDOT:PSS ICL. **d** For a C$_{60}$/SALD-SnO$_2$/Au/PEDOT:PSS ICL. The $J_{sc}$ was obtained by integrating with the AM1.5G spectrum. **b** Fast dark and light $J$–$V$ scans of tandem cells with different ICLs (measured with 6.76 mm$^2$ aperture area). Source data are provided as a Source Data file.

after 300 s of steady-state power output tracking further confirms the device performance (Fig. 3e). The device produces a stabilized $V_{oc}$ of 1.89 V, which is very close to the summed $V_{oc}$ value of the 1.73 eV front sub-cell (1.13 V) and 1.23 eV back sub-cell (0.78 V). This low potential energy loss, together with a high FF of 0.81, implies the fast recombination of charges from adjacent sub-cells in the ICLs[27]. As discussed, our tandem cell performance is limited by the low $J_{sc}$. Compared to a $J_{sc}$ of 28.0 mA cm$^{-2}$ obtained for 1.23 eV single-junction PSC, the summed EQE spectrum of both sub-cells is generally lower in the near-infrared range and shows a current density of 26.8 mA cm$^{-2}$ (Supplementary Fig. 13). Here, the loss in $J_{sc}$ is mainly attributed to parasitic absorption from the ICL (PEDOT:PSS) and ITO substrate, reflectance, optical interference, and insufficient 1.23 eV absorber layer thickness for light absorption (Supplementary Fig. 14).

**Monolithic all-perovskite triple-junction solar cells.** Further-more, we integrated our two-step processed 1.73 eV, 1.57 eV, and 1.23 eV perovskite absorbers into monolithic all-perovskite triple-junction solar cells. Similar to tandem cells, our initial test found that the triple-junction device with PCBM/SALD-SnO$_2$/PEDOT:PSS ICLs displays an s-kink in the $J$–$V$ characteristics, which can be removed by replacing PCBM with C$_{60}$ and inserting a thin Au layer at the SALD-SnO$_2$/PEDOT:PSS interface (Supplementary Fig. 15). As shown in Fig. 4a, b, the optimized triple-junction cell utilized a device configuration of ITO/PTAA/Cs$_{0.1}$(FA$_{0.66}$MA$_{0.34}$)$_{0.9}$PbI$_2$Br/C$_{60}$/ SALD-SnO$_2$/Au/PEDOT:PSS/PTAA/FA$_{0.66}$MA$_{0.34}$PbI$_{2.85}$Br$_{0.15}$/C$_{60}$/ SALD-SnO$_2$/Au/PEDOT:PSS/FA$_{0.66}$MA$_{0.34}$Pb$_{0.5}$Sn$_{0.5}$I$_3$/C$_{60}$/BCP/Ag. In such a device stack, the 1.73-eV perovskite absorber was further

reduced to ~100 nm to approach a current matching condition (Supplementary Fig. 6), and PTAA was deposited on PEDOT:PSS in the 1.57 eV middle sub-cell to achieve better device performance (Supplementary Fig. 16). In the stabilized $J$–$V$ measurement, the best-performing triple device exhibits a PCE of 16.8% (6.76 mm$^2$ aperture area), with a $J_{sc}$ of 7.4 mA cm$^{-2}$, a $V_{oc}$ of 2.78 V, and an FF of 0.81. The triple cell performance is higher than that of single-junction PSCs prepared in the same batch (Fig. 4c and Table 3). We also note that the thermal stressing imposed on the 1.73 and 1.57 eV sub-cells during the fabrication of a triple-junction cell should not affect their device performance (Supplementary Fig. 17). The corresponding EQE spectra generate photocurrents of 8.2, 8.9, and 7.6 mA cm$^{-2}$ for the 1.73 eV front sub-cell, 1.57 eV middle sub-cell, and 1.23 eV back sub-cell, respectively (Fig. 4d), indicating that the narrow bandgap perovskite sub-cell is limiting the $J_{sc}$ of the triple device. Also, the device shows negligible hysteresis between reverse, forward, and stabilized $J$–$V$ characteristics (Fig. 4e). The PV performance of the triple-junction cell is further confirmed by a stabilized PCE of 16.9% after 300 s of steady-state power output tracking (Fig. 4f). Meanwhile, the stabilized $V_{oc}$ (2.78 V) of the triple device is close to the summed $V_{oc}$ value (2.86 V) of the corresponding single-junction PSCs, sug-gesting the effectiveness of such ICLs in our triple-junction design. Furthermore, a statistical summary of eight triple cells illustrates a narrow distribution of PCEs, which demonstrates the good repro-ducibility of our approach to fabricate triple-junction cells (Fig. 4g and Supplementary Table 4).

Under the current design, the PCE of our triple-junction solar cell is limited by the $J_{sc}$ of 1.23 eV back sub-cell. In contrast to the 1.23 eV single-junction cell (27.0 mA cm$^{-2}$), the summed EQE of all three sub-cells in a triple device (24.7 mA cm$^{-2}$) suggests a

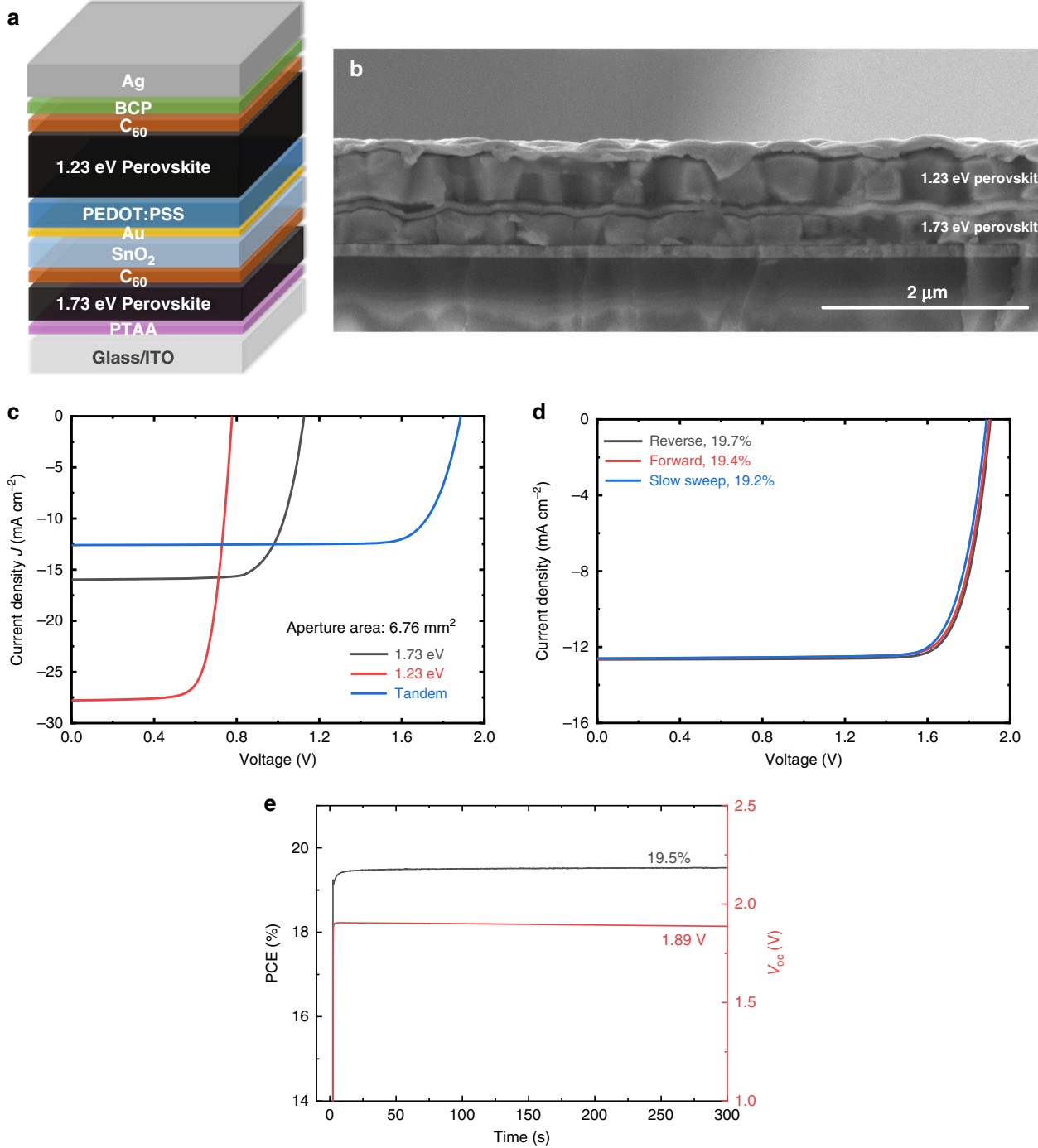

**Fig. 3 Device configuration and performance of optimized monolithic all-perovskite tandem solar cells. a** Device structure. **b** Corresponding cross-sectional SEM image of the tandem device. Scale bar is 2 μm. **c** Stabilized *J–V* curves of the best-performing tandem cell and the corresponding 1.73 eV and 1.23 eV single-junction PSCs prepared in the same batch (measured with 6.76 mm² aperture area). **d** Reverse, forward, and stabilized *J–V* scans. **e** PCE and $V_{oc}$ tracking of the best-performing tandem solar cell. Source data are provided as a Source Data file.

**Table 2 Photovoltaic parameters of the single-junction and tandem PSCs.**

| Devices | $J_{sc}$ (mA cm$^{-2}$) | $V_{oc}$ (V) | FF | PCE[a] (%) | $J_{sc}$[b] (mA cm$^{-2}$) | PCE[c] (%) |
|---|---|---|---|---|---|---|
| 1.73 eV (300 nm) | 16.0 | 1.13 | 0.73 | 13.2 | 15.5 | 12.7 |
| 1.23 eV | 27.8 | 0.78 | 0.73 | 15.8 | 28.0 | 15.9 |
| Tandem | 12.6 | 1.89 | 0.81 | 19.2 | – | – |

[a]The data was extracted from stabilized *J–V* curves under simulated AM 1.5G illumination (100 mW cm$^{-2}$). The aperture area was 6.76 mm².
[b]Calculated by integrating the EQE spectrum with the AM1.5G spectrum.
[c]Corrected PCE obtained by calculating the $J_{sc}$ integrated from EQE spectrum and $V_{oc}$ and FF from the stabilized *J–V* measurement.

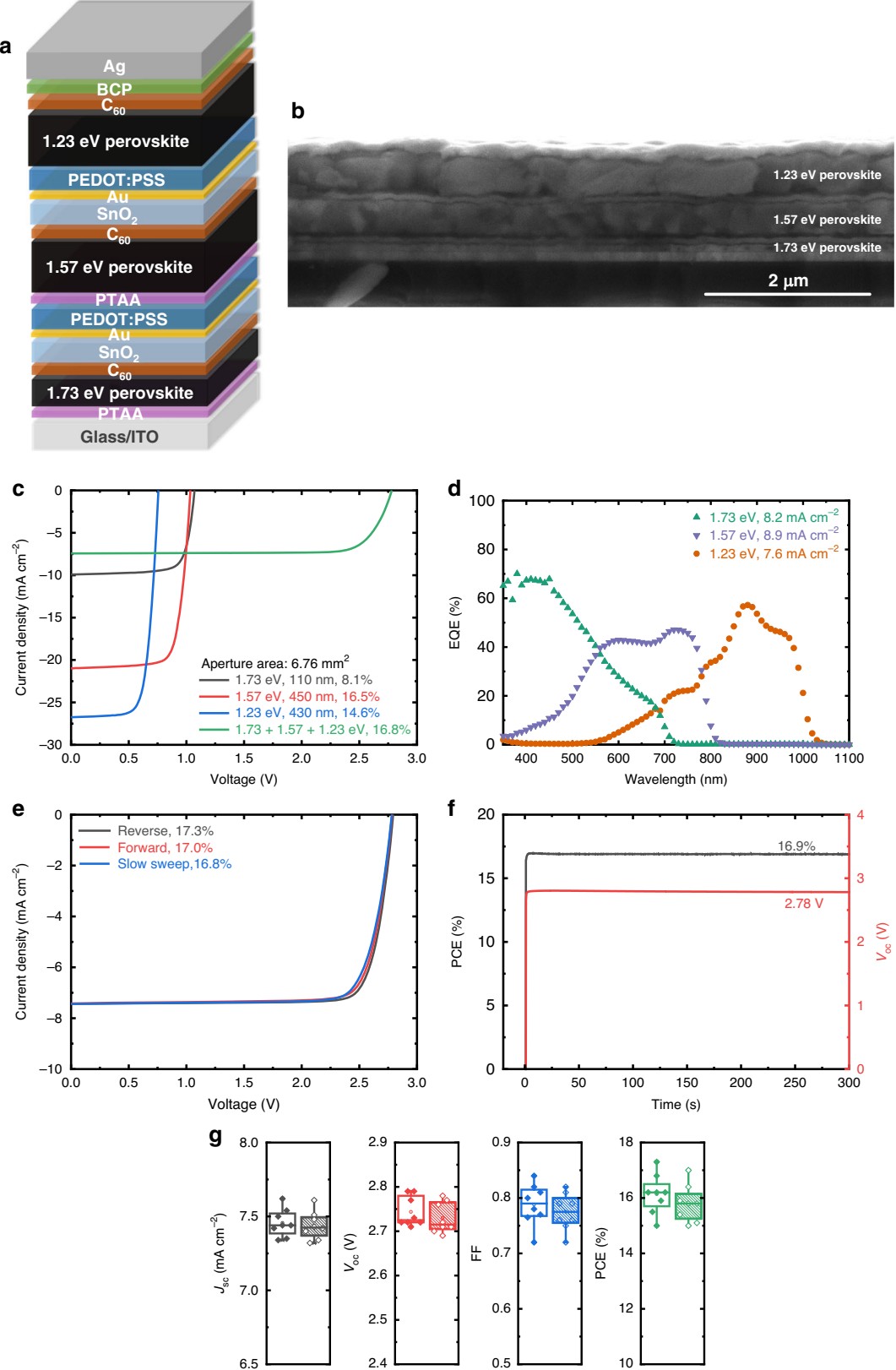

**Fig. 4 Device structure and PV performance of the optimized monolithic triple-junction solar cell.** Device configuration (**a**) and the corresponding cross-sectional SEM image (**b**) of the triple cell. Scale bar is 2 μm. **c** Stabilized J–V curves of the best-performing triple device and the corresponding 1.73 eV, 1.57 eV, and 1.23 eV single-junction PSCs prepared in the same batch (6.76 mm² aperture area). **d** EQE spectra of 1.73 eV, 1.57 eV, and 1.23 eV sub-cells in a triple-junction device with $C_{60}$/SALD-$SnO_2$/Au/PEDOT:PSS ICLs, the $J_{sc}$ was obtained by integrating with the AM 1.5G spectrum. Reverse, forward, and stabilized J–V scans (**e**) and PCE and $V_{oc}$ tracking (**e**) of the best-performing triple device. **g** Statistical distributions of PV parameters of $J_{sc}$, $V_{oc}$, FF, and PCE of triple-junction solar cells, measured under fast J–V sweep in reverse (left) and forward (right) directions. In the boxplots the mean (open circle), median (center line), 25th and 75th percentiles (box limits), and 5th and 95th percentiles (whiskers) are shown. Source data are provided as a Source Data file.

**Table 3 Photovoltaic parameters of the single-junction and triple-junction PSCs.**

| Devices | $J_{sc}$ (mA cm$^{-2}$) | $V_{oc}$ (V) | FF | PCE[a] (%) | $J_{sc}$[b] (mA cm$^{-2}$) | PCE[c] (%) |
|---|---|---|---|---|---|---|
| 1.73 eV (100 nm) | 9.9 | 1.07 | 0.76 | 8.1 | 10.7 | 8.7 |
| 1.57 eV | 21.0 | 1.03 | 0.76 | 16.5 | 20.4 | 16.0 |
| 1.23 eV | 26.7 | 0.76 | 0.72 | 14.6 | 27.0 | 14.8 |
| Triple | 7.4 | 2.78 | 0.81 | 16.8 | – | – |

[a]The data were extracted from stabilized $J$–$V$ curves under simulated AM 1.5G illumination (100 mW cm$^{-2}$). The aperture area is 6.76 mm$^2$.
[b]Calculated by integrating the EQE spectrum with the AM1.5G spectrum.
[c]Corrected PCE obtained by calculating the $J_{sc}$ integrated from EQE spectrum and $V_{oc}$ and FF from the stabilized $J$–$V$ measurement.

considerable loss in the near-infrared, which accounts for a loss in photocurrent of 2.3 mA cm$^{-2}$ (Supplementary Fig. 18). Similar to the tandem analysis, the $J_{sc}$ loss is mainly originated from parasitic absorption of ITO substrate and PEDOT:PSS, reflection, optical interference, and also incomplete light absorption in the near-infrared due to a relatively thin 1.23 eV absorber layer (Supplementary Fig. 19). As previously discussed by Snaith and co-workers[10,22], further improving the PCE of all-perovskite triple-junction solar cells would require an efficient ~2 eV wide bandgap perovskite, which enables a more balanced light absorption in each absorber layer and provides a much higher $V_{oc}$ in the triple-junction cell. However, such wide bandgap PSCs with low $V_{oc}$ deficit has not been reported to date (Supplementary Figs. 21 and 22 and Supplementary Table 5). Development of wide bandgap perovskite materials with a small $V_{oc}$ deficit ($E_g/q - V_{oc}$) and good photo-stability would significantly advance high-efficiency all-perovskite triple-junction solar cells.

## Discussion

In summary, we have reported a universal two-step solution process to fabricate PSCs with bandgaps of 1.73 eV, 1.57 eV, and 1.23 eV. By optimizing solvent-robust and low-resistivity ICLs, PCEs of above 19% are achieved in monolithic all-perovskite tandem solar cells with 1.73 eV and 1.23 eV absorbers. Furthermore, our strategy enables us to fabricate efficient and reproducible monolithic all-perovskite triple-junction solar cells. A triple device comprising a 1.73 eV, 1.57 eV, and 1.23 eV sub-cells shows a promising PCE of 16.8%, with very low potential energy drop of only 80 mV in comparison to the summed $V_{oc}$ of all three sub-cells ($V_{oc} = 2.78$ V) and low-resistivity loss (FF = 0.81). Further improving the performance of ~2 eV wide bandgap perovskite, will be essential for the future development of high-efficiency all-perovskite triple-junction solar cells. During the submission process, we became aware of a related work on multi-junction perovskite solar cells[42].

## Methods

**Preparation of perovskite precursor solutions**. All materials were purchased from commercial sources and used as received unless stated otherwise. For 1.73 eV $Cs_{0.1}(FA_{0.66}MA_{0.34})_{0.9}PbI_2Br$, 876 μL DMF (99.8%) and 86.4 μL DMSO (99.9%) were sequentially added to 34.8 mg CsI (Sigma-Aldrich, 99.999%) and 553 mg PbI$_2$ (Sigma-Aldrich, 99.999%) to form a ~1.25-M precursor solution, before stirring at 60 °C overnight. The small portion of DMSO was added to form a 1:1 molar ratio of PbI$_2$:DMSO. The order of solvent addition does not change the device performance. In all, 34.9 mg FABr (Greatcell Solar) and 16.0 mg MABr (Greatcell Solar) were dissolved in 1 ml isopropanol (99.5%; 0.422 M). Thinner $Cs_{0.1}(FA_{0.66}$-$MA_{0.34})_{0.9}PbI_2Br$ films were prepared by reducing the precursor concentrations. For a 300-nm-thick layer, 1.1 M CsI-PbI$_2$ and 0.297 M FABr-MABr solutions were used. For a 100-nm-thick layer, 0.4 M CsI-PbI$_2$ and 0.095 M FABr-MABr solutions were prepared. For the 1.57 eV $FA_{0.66}MA_{0.34}PbI_{2.85}Br_{0.15}$ precursor solutions, 553 mg PbI$_2$ was dissolved in 876 μL DMF and 86.4 μL DMSO. 54.0 mg FAI (Greatcell Solar), 14.3 mg MAI (Greatcell Solar), and 7.6 mg MABr (Greatcell Solar) were dissolved in 1 ml isopropanol (0.471 M). All the solutions above were kept at 60 °C overnight. For the 1.23 eV $FA_{0.66}MA_{0.34}Pb_{0.5}Sn_{0.5}I_3$ precursor solutions, 276.5 mg PbI$_2$ and 223.4 mg SnI$_2$ (Sigma-Aldrich, 99.99%) mixture were dissolved in 876 μL DMF and 86.4 μL DMSO, and 10 mol% SnF$_2$ (Sigma-Aldrich, 99%) was added with respect to SnI$_2$. The solution was stirred at 60 °C for 1 h and filtered by a PTFE syringe filter (0.22 μm). In all, 53.5 mg FAI and 25.6 mg MAI mixture were dissolved in 1 mL of

isopropanol and stirred at 60 °C for 1 h. We note that all materials except PEDOT: PSS and PTAA (stored in ambient) were stored in a dry N$_2$ glovebox. While it was crucial to weigh Sn-containing compounds in a dry N$_2$ glovebox, other compounds showed little influence on the device performance when weighed either in dry N$_2$ or in an ambient atmosphere. All solvents for spin coating were stored in an N$_2$ glovebox. All solutions were prepared and stirred in the same glovebox and were cooled to room temperature before use.

**Device fabrication**. Pre-patterned ITO glass substrates (Naranjo, 17 Ω/sq, the substrate layout is shown in Supplementary Fig. 23) were cleaned by sonication in acetone, sodium dodecyl sulfate (Acros, 99%), soapy water, deionized water, and isopropanol (15 min). Before spin coating, the ITO substrates were further treated under UV-ozone for 30 min. For both the 1.73-eV wide bandgap and 1.57-eV mid bandgap PSCs, 3 mg ml$^{-1}$ of PTAA in toluene (TCI, 99.5%) was spin-coated onto the ITO substrate at 5700 rpm for 30 s and annealed at 100 °C for 10 min in the N$_2$ glovebox. After cooling down, the Pb-containing precursor solution was spin-coated at 3000 rpm (with a 2000 rpm/s acceleration) for 30 s, followed by the dynamic spin-coating of FA/MA-based precursor solution at a speed of 3000 rpm for 30 s. The substrate was immediately annealed at 100 °C for 30 min. For the devices using PCBM as the electron transport layer (ETL), 20 mg mL$^{-1}$ of PCBM (Solenne BV, 99%) in 1 mL chlorobenzene (CB, 99.8%) and chloroform (CF, 99%) mixture (1:1 volume ratio) was spin-coated at 1000 rpm for 60 s, followed by thermal annealing at 100 °C for 30 min. Finally, 1 nm LiF (0.2 Å/s) and 100 nm Al (2 Å/s) were thermally evaporated to complete the device fabrication. For devices with C$_{60}$ (SES Research, 99.95%) as the ETL, 20 nm C$_{60}$ (0.5 Å/s), 8 nm BCP (Lumtec, 99%) (0.5 Å/s), and 100 nm Al (2 Å/s) were thermally evaporated in a sequence. A similar approach was used for 1.23 eV narrow bandgap PSCs. PEDOT: PSS (Heraeus Clevios, PCP Al 4083) was filtered by a PVDF filter (0.45 μm), spin-coated on the cleaned ITO substrate at 3000 rpm for 60 s and annealed at 140 °C for 15 min in ambient condition. In the N$_2$ glovebox, the PbI$_2$/SnI$_2$ precursor solution was spin-coated at 3000 rpm (with a 2000 rpm acceleration) for 30 s. After drying the wet precursor film at room temperature for 30 min, FAI/MAI precursor solution was dynamically spin-coated on top at 3000 rpm for 60 s, followed by thermal annealing at 100 °C for 30 min. Afterward, 1 mg mL$^{-1}$ NH$_4$SCN (Sigma-Aldrich, 97.5%) dissolved in isopropanol was spin-coated on top at 5000 rpm for 30 s. Finally, 20 nm C$_{60}$ (0.5 Å/s), 8 nm BCP (0.5 Å/s), and 100 nm Ag (2 Å/s) were sequentially evaporated on top. The cell area was determined by the overlap of the top and bottom ITO electrodes (0.09 or 0.16 cm$^2$). For tandem and triple devices, the same procedure was used to fabricate different perovskite sub-cells. Between 1.73/1.57 eV sub-cells and 1.57/1.23 eV sub-cells, after evaporation of 20 nm C$_{60}$, the samples were transferred in the air to a homemade spatial ALD reactor as described previously[43]. Tetrakis(dimethylamino)tin(IV) bubbler was used as tin source while H$_2$O was used as co-reactant. Both vessels were kept at room temperature flowing 500 sscm of argon through them. The process was carried out at 100 °C having nominal growth per cycle GPC of 0.125 nm/cycle, determined on silicon wafer. The final thickness of the layer was 45 nm. The substrates were then transferred back to the thermal evaporator for the deposition of a 1-nm Au layer. For Au evaporation, a shadow mask with aperture slightly larger than the cell area was applied. The samples were then exposed in air to spin-coat PEDOT:PSS at a speed of 3000 rpm for 60 s, and annealed at 100 °C for 15 min.

**Device characterization**. The $J$–$V$ and EQE characteristics were performed in the N$_2$ glovebox at room temperature. A tungsten-halogen lamp combined with a UV-filter (Schott GG385) and a daylight filter (Hoya LB120) was used to simulate the solar spectrum, the light intensity was calibrated by Si photodiode to be ~100 mW cm$^{-2}$. A black shadow mask with an aperture slightly smaller than the cell area was used (0.0676 or 0.1296 cm$^2$). For the fast $J$–$V$ measurements, a Keithley 2400 source meter was used to sweep the voltage from +1.5 V (–0.5 V) to –0.5 V (+1.5 V) at a scan rate of 0.25 V s$^{-1}$ in reverse (forward) scan. For the stabilized $J$–$V$ tests, the solar cell was first monitored at $V_{oc}$ for 5 min, followed by a reverse voltage sweep from ($V_{oc}$ + 0.02) V to –0.02 V at a step size of 0.02 V. During the voltage sweep, a Keithley measures the current density for 5 s at each voltage point. From the stabilized $J$–$V$ curves, the voltage at the maximum power point was extracted and was applied to the cell during steady-state power output tracking. In the EQE measurements, a modulated (Oriel, Cornerstone 130)

tungsten-halogen lamp (Philips focusline, 50 W) was used as the light source. The signal of solar cells was amplified by a current preamplifier (Stanford Research, SR 570) and measured by a lock-in amplifier (Stanford Research, SR 830). The spectral response was then transformed into EQE using a calibrated silicon reference cell. For single-junction PSCs, to mimic the one-sun condition for the $J$–$V$ measurements, additional LED bias light (530 nm for 1.73/1.57 eV, 940 nm for 1.23 eV, Thorlabs) was used to generate a photocurrent close to $J_{sc}$ in the cell during the EQE measurement. We note that for our single-junction PSCs, the difference between non-biased and light-biased EQE spectra is negligible. For tandem solar cells, a 530-nm bias light was used to measure the EQE response of 1.23 eV back sub-cell, whereas a 940-nm bias light was used for the EQE of 1.73 eV front sub-cell. In a triple-junction solar cell, the 530 nm bias light was used for the 1.23 eV back sub-cell, the 730-nm bias light was used for the 1.73 eV front sub-cell, and a combination of 530 + 940 nm bias light was used to measure the 1.57 eV middle sub-cell. We also studied the effect of voltage bias by applying the sum of the $V_{oc}$ of the optically biased sub-cells on our triple-junction solar cells during the EQE measurements and found that the difference between voltage biased and non-voltage biased EQE spectra is insignificant.

**Film characterization**. SEM images were recorded by a FEI Quanta 3D FEG microscope, using a 5-kV electron beam and a secondary electron detector. XRD patterns were obtained by a Bruker 2D phaser (Cu Kα radiation, $\lambda = 1.5405$ Å). UV-vis-NIR absorption measurements were performed by PerkinElmer Lambda 900 UV-vis-NIR spectrophotometer. Steady-state photoluminescence spectra were measured by Edinburgh Instruments FLSP920 double-monochromator luminescence spectrometer, with a near-infrared photomultiplier (Hamamatsu).

**Simulations**. Optical modeling was carried out with the GenPro4 program[44].To ensure the reliability of any conclusions drawn from optical modeling, component materials of the semi-transparent PSCs were prepared individually on glass and characterized with a J.A. Woollam ellipsometer to acquire optical constants. Furthermore, measured R and T of each material were compared to those simulated with its optical constants for validation.

**Reporting summary**. Further information on research design is available in the Nature Research Reporting Summary linked to this article.

## Data availability
All relevant data in this study are available from the corresponding author upon request. Source data are provided with this paper.

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

## Acknowledgements

We thank Veronique Gevaerts, Herbert Lifka, and Francesco Di Giacomo for initial discussions leading to this work and Adriana Creatore for advice on ALD. The research has received funding from the Netherlands Organisation for Scientific Research via the NWO Spinoza grant awarded to R.A.J. Janssen and the Joint Solar Programme III (project 680.91.011). We further acknowledge funding from the Ministry of Education, Culture, and Science (Gravity program 024.001.035) and from the Dutch Ministry of Economic Affairs and the Netherlands Enterprise Agency (RVO) for funding via the Top-consortia Knowledge and Innovation (TKI) Program in project "ALLSUN" (TEZ0214010).

## Author contributions

J.W. and V.Z. manufactured the multijunction cells. J.W. and K.D. developed the sub cells. D.Z. performed the optical simulations. Characterization of materials and devices was done by J.W., while K.D. performed XRD experiments. J.W., V.Z., M.M.W. and R.A. J.J. planned the research and analysed the results; J.W. wrote the manuscript, all authors commented on it. R.A.J.J. supervised the project.

## Competing interests

The authors declare no competing interests.
