## [Peer Review File · Nature Communications]

REVIEWER COMMENTS

Reviewer #1 (Remarks to the Author):

The main novelty of this paper lies in the demonstration of a triple-junction all perovskite device of reasonable efficiency (16.8%). This is a significant improvement upon the only other available demonstration of 6.7%. Their realisation of a high quality interlayer which nearly eliminates electronic loss is also very valuable. The paper does not skimp on detail, making it convenient for other researchers to reproduce. In short, this is a fantastic paper which in my view should be published in Nature Communications. There are few minor issues which should be addressed before publication:

1. "Previously, it has been demonstrated that a SnO₂ layer grown by atomic layer deposition (ALD) can prevent solvent damage without using the sputtered indium tin oxide (ITO)." It would be constructive to compare the recombination layers in the literature, explaining the different sputtered layers which have been used, to the recent ALD SnO₂ development, with and without ITO. I am not asking for more experiments, just a bit of an expansion on the discussion of this point. It is very interesting that SnO₂ + very thin (non continuous) Au works, but optically what is best?

2. Explain advantages of S-ALD compared to conventional thermal ALD/PEALD eg. Reduction of ALD cycle time, potential processing at lower temperatures. Refer to use of S-ALD in literature for perovskite devices. Examples below:

L. Hoffmann et al., "Spatial Atmospheric Pressure Atomic Layer Deposition of Tin Oxide as an Impermeable Electron Extraction Layer for Perovskite Solar Cells with Enhanced Thermal Stability", ACS Appl. Mater. Int., 10, 6006-6013, 2018

M. Najafi et al., "Highly Efficient and Stable Semi-Transparent p-i-n Planar Perovskite Solar Cells by Atmospheric Pressure Spatial Atomic Layer Deposited ZnO," Sol. RRL, vol. 1800147, 1800147, 2018

V. Zardetto et al., "Towards Large Area Stable Perovskite Solar Cells and Modules," IEEE PVSC, 978-1-7281-0494-2, 2019

3. Clearly state the cell aperture area (0.09 cm² versus 0.16 cm²) for the PCE achieved for single junction, tandem and triple junction devices. Also provide your cell layout with dimensions in the supplementary. This would be important for researchers trying to reproduce your high fill factors.

4. Include J-V curves in the supplementary information showing the increased device performance achieved with PTAA deposited on top of PEDOT:PSS for the 1.57 eV perovskite absorber device versus PTAA.

5. It would be useful to know the reasoning for the DMF:DMSO volume ratio of ~10.1:1?

6. In experimental methods section it is important to provide the growth parameters and details for S-ALD SnO₂:

Deposition temperature

Exposure time of precursors

GPC, number of ALD cycles per rotation of substrate, substrate velocity/rotation frequency
What carrier gas was utilised and it's flow rate
At what temperature were precursor vessels held, was a bubbler utilized and the precursor partial pressures

7. Please mention the sheet resistance of the ITO used.

8. Spectral mismatch corrected PCE has to be reported. Spectral mismatch can cause significant overestimation of PCE in multi-junctions. The authors could follow the approach shown by McMeekin, D. P. et al. Solution-Processed All-Perovskite Multi-junction Solar Cells. *Joule* 3, 387–401 (2019) to estimate the mismatch of the current limiting junction.

9. Clarification is needed on whether solutions were weighed and stirred in the glovebox. Giving precise details such as these really help other researchers in the field to follow the work.

10. "A triple device comprising a 1.73 eV, 1.57 eV, and 1.23 eV sub-cells shows a promising PCE of 16.8%, with very low potential energy drop ($V_{oc} = 2.78$ V) and resistivity loss ($FF = 0.81$)" It is suggested that the actual potential drop (70 mV) be mentioned; something like: "A triple device comprising a 1.73 eV, 1.57 eV, and 1.23 eV sub-cells shows a promising PCE of 16.8%, with very low potential energy drop of only 70mV in comparison to the sum of the V_{oc} of each sub cell ($V_{oc} = 2.78$ V) and low resistivity loss ($FF = 0.81$)"

11. Simulation: Mention all the layer thicknesses used, and report the n,k values where measured. If taken from literature, provide the appropriate references.

Reviewer #2 (Remarks to the Author):

In this work, Wang et al. reported the fabrication of all-perovskite tandem and triple-junction solar cells, where the perovskite layers were all fabricated with two-step sequential deposition method. The topic of all-perovskite multijunction solar cells is of high interests now for the PV community, since it offers potential to obtain higher PCEs than single-junction perovskite solar cells.

The processing of lead halide perovskites using two-step sequential method has been widely reported and investigated in the community. I do not see much new ideas herein. The processing of mixed Sn-Pb narrow-bandgap perovskite has been, however, rarely reported. Compared to authors' recent work on two-step processed Sn-Pb perovskite solar cells (DOI: 10.1002/aenm.202000566), I did not find any advances in either performance or processing approach itself. The design of interlayer using ALD deposited SnO₂ and evaporated Au has also been reported in previous work (10.1038/s41560-019-0466-3). For the tandem cells herein, the demonstrated PCEs in this work are much lower than those of previous works (e.g. 10.1038/s41560-019-0466-3; 10.1126/science.aav7911 and others). I would be happy to reconsider this work, if the authors could demonstrate PCEs of Sn-Pb single junction cell and tandem cells comparable or even higher than previous works.

Other comments:

The author chose FA/MA as the main composition for the wide-bandgap and mid-bandgap perovskites. Would following processing affect the performance of MA-containing materials? It seems that MA-incorporated perovskites are not very stable under thermal stressing. Another issue is the photo-induced instability of MA-contained wide-bandgap perovskites.

The bandgap (~1.73 eV) of wide-bandgap subcell used for the triple-junction is far from ideal situation (~2.0 eV). Authors should consider fabricating triple-junction cells with better bandgap matching. Otherwise, it loses the benefits of making triple-junction configuration.

Reviewer #3 (Remarks to the Author):

This manuscript is well written and has demonstrated some novelty in the universal two-step preparation methods for triple-junction all-perovskite tandem solar cells. Overall this is nice work, though PCE is relatively. I only have two minor technical comments:

[1] On Page 9, the authors attribute the reduced J_{sc} (from 14 to 13.1 mA/cm²) in the 1.23-eV subcell of the C60-based tandem devices is attributed to the change in optical interference. Do the changes in the EQE interference agree with the simulation?

[2] On Page 13, PTAA was deposited on PEDOT:PSS in the 1.57 eV mid subcell. Did the authors test the direct replacement of PEDOT:PSS by PTAA for the mid subcell?

We thank the reviewers for their valuable comments and effort to improve our manuscript. Below we repeat the questions and concerns of the reviewers (in red font) and include our response and outline the changes made in the revised manuscript (highlighted).

Reviewer #1 (Remarks to the Author):

1. “Previously, it has been demonstrated that a SnO₂ layer grown by atomic layer deposition (ALD) can prevent solvent damage without using the sputtered indium tin oxide (ITO).”

It would be constructive to compare the recombination layers in the literature, explaining the different sputtered layers which have been used, to the recent ALD SnO₂ development, with and without ITO. I am not asking for more experiments, just a bit of an expansion on the discussion of this point. It is very interesting that SnO₂ + very thin (non continuous) Au works, but optically what is best?

Response:

Following the reviewer’s advice, we expanded our discussion on the development of interconnecting layers for all-perovskite based tandem in the Introduction part (on Page 3-4):

For all-perovskite multijunction cells, examples of solution-processed ICLs are p-doped cross-linked poly(triarylamine) (PTAA)/n-doped phenyl-C₆₁-butyric acid methyl ester (PCBM)^[1] and poly(3,4-ethylene dioxythiophene):polystyrene sulfonate (PEDOT:PSS)/indium tin oxide (ITO) nanoparticles^[2]. In comparison, using sputtered ITO and indium zinc oxide (IZO) as the recombination layers has enabled higher efficiency in tandem devices^[3-6], albeit the increased lateral shunt pathways and optical losses in the near-infrared region^[4]. Moreover, a thin metal oxide layer such as SnO₂ and Al-doped ZnO (AZO) prepared by atomic layer deposition (ALD) was necessary to prevent sputter damage^[3,5-7]. It has been previously demonstrated that tuning the growth conditions can yield compact and conductive ALD layers, which alone prevent solvent damage and allow for fast charge transport after depositing a thin TCO layer^[5,8-9]. Recently, Tan et al. utilized TCO-free ICLs based on C₆₀/ALD-SnO₂/Au/PEDOT:PSS for all-perovskite tandem solar cells^[10].

Indeed, despite the improvement in the FF, the extra Au layer can induce considerable parasitic absorption loss in the multijunction cells. As we discussed in the manuscript, the resistivity-loss originates from the low carrier density of the SnO₂ layer. Introducing ozone (Behrendt et al. *Adv. Mater.* 2015, 27, 5961–5967) or O₂ plasma (Kuang et al. *ACS Appl. Mater. Interfaces* 2018, 10, 30367–30378) at a low-temperature ALD process is likely to improve the electron mobility of the SnO₂ layer, provided that its optical transparency is still suitable for multijunction applications. Alternatively, the Au layer can be replaced by a conductive and more optically transparent interlayer, e.g., ITO, which has a refraction index similar to that of SnO₂.

2. Explain advantages of S-ALD compared to conventional thermal ALD/PEALD eg. Reduction of ALD cycle time, potential processing at lower temperatures. Refer to use of S-ALD in literature for perovskite devices. Examples below:

L. Hoffmann et al., “Spatial Atmospheric Pressure Atomic Layer Deposition of Tin Oxide as an Impermeable Electron Extraction Layer for Perovskite Solar Cells with Enhanced Thermal Stability”, *ACS Appl. Mater. Int.*, 10, 6006-6013, 2018

M. Najafi et al., “Highly Efficient and Stable Semi-Transparent p-i-n Planar Perovskite Solar Cells by Atmospheric Pressure Spatial Atomic Layer Deposited ZnO,” *Sol. RRL*, vol. 1800147,

1800147, 2018

V. Zardetto et al., "Towards Large Area Stable Perovskite Solar Cells and Modules," IEEE PVSC, 978-1-7281-0494-2, 2019

Response:

Following the reviewer's suggestion, in the Introduction (Page 4) we expanded our discussion on S-ALD technique and further referred to the references as suggested:

Nevertheless, conventional ALD technique requires vacuum and is limited by a low deposition rate. Atmospheric pressure spatial-ALD (SALD) can be done at a much higher deposition rate while preserving conformal and pinhole-free depositions, which is closer to the industrial manufacturing requirements³⁵⁻³⁸.

3. Clearly state the cell aperture area (0.09 cm² versus 0.16 cm²) for the PCE achieved for single junction, tandem and triple junction devices. Also provide your cell layout with dimensions in the supplementary. This would be important for researchers trying to reproduce your high fill factors.

Response:

Following the reviewer's suggestion, we added the substrate layout (ABAB) used in this study in Supplementary Fig. 23 (Page S15, Supporting Information), with further explanations:

Devices were made on 30 × 30 mm² substrates with patterned ITO (17 Ohm/sq). Device areas were determined by the overlap of the ITO stripes (lighter grey) with the patterned metal back contacts (darker grey), providing cell areas of 3.0 × 3.0 mm² and 4.0 × 4.0 mm². Extra metal contacts were evaporated onto the cleaned ITO contacts to reduce the device series resistance through the ITO. For $J-V$ measurements, cells were illuminated through square masks of 2.6 × 2.6 mm² and 3.6 × 3.6 mm² for the 9.0 and 16.0 mm² size cells, respectively. Each cell during the $J-V$ measurement was manually fixed to the same position of our setup, to avoid any differences due to spatial non-uniformity of the light source.

We further clarified the cell aperture area when reporting the $J-V$ curves or PCEs in the manuscript, i.e., Page 6 (Figure 1f), 7 (Table 1), 9 (Figure 2b), 11 (Figure 3c), 11 (Table 2), 13, 14 (Figure 4c), and 14 (Table 3); Supplementary Figure 5, 6, 10, 11, 15 and Table S2, S3, and S4.

4. Include J-V curves in the supplementary Information showing the increased device performance achieved with PTAA deposited on top of PEDOT:PSS for the 1.57 eV perovskite absorber device versus PTAA.

Response:

Following the reviewer's suggestion, we have now added the device performance of the 1.57 eV single-junction cells with different bottom contacts (Supplementary Fig. 16): (a) ITO/PEDOT:PSS (and/or PTAA), (b) ITO/SALD-SnO₂/PEDOT:PSS (and/or PTAA), and (c) ITO/SALD-SnO₂/Au/PEDOT:PSS (and/or PTAA) in Supplementary Fig. 16, page S11. These demonstrate that the ICL combination of SALD-SnO₂/(Au)/PEDOT:PSS/PTAA outperforms both SALD-SnO₂/(Au)/PEDOT:PSS and SALD-SnO₂/(Au)/PTAA in the 1.57 eV solar cells.

5. It would be useful to know the reasoning for the DMF:DMSO volume ratio of ~10.1:1?

Response:

As we know, one challenge of the two-step method is to fabricate perovskite films without unconverted PbI_2 . The solvation state of PbI_2 can drastically change the crystallization dynamics of perovskite films. It has been previously reported that a $\text{PbI}_2(\text{DMSO})_1$ complex precursor enables a fast and complete conversion into the perovskite phase (Seok et al. Science, 2015, 6240, 1230-1234.)

In this study, we use $\sim 1.25 \text{ mol L}^{-1}$ of PbI_2 (or $\text{Pb}_{0.5}\text{Sn}_{0.5}\text{I}_2$) precursor solution in the first step. For 553 mg PbI_2 , 1:1 molar ratio of PbI_2 :DMSO was formed by adding $\sim 86 \mu\text{L}$ DMSO, then $\sim 876 \mu\text{L}$ DMF was used to develop a total solution concentration of $\sim 1.25 \text{ M}$. Therefore, the volume ratio of DMF:DMSO is close to 10.1:1.

To clarify this, we added a note in the Methods section (Page 15): **The small portion of DMSO was added to form a 1:1 molar ratio of PbI_2 :DMSO.**

6. In experimental methods section it is important to provide the growth parameters and details for S-ALD SnO_2 :

Deposition temperature

Exposure time of precursors

GPC, number of ALD cycles per rotation of substrate, substrate velocity/rotation frequency

What carrier gas was utilised and it's flow rate

At what temperature were precursor vessels held, was a bubbler utilized and the precursor partial pressures

Response:

We have added the following information in the Methods (Page 17):

Tetrakis(dimethylamino)tin(IV) bubbler was used as tin source while H_2O was used as co-reactant. Both vessels were kept at room temperature flowing 500sccm of argon through them. The process was carried out at 100°C having nominal growth per cycle GPC of 0.125nm/cycle , determined on silicon wafer. The final thickness of the layer was 45 nm .

7. Please mention the sheet resistance of the ITO used.

Response:

The sheet resistance of our ITO substrate (ABAB layout) is **$17 \Omega/\text{sq}$** . We added this Information to the Methods section (Page 16).

8. Spectral mismatch corrected PCE has to be reported. Spectral mismatch can cause significant overestimation of PCE in multi-junctions. The authors could follow the approach shown by McMeekin, D. P. et al. Solution-Processed All-Perovskite Multi-junction Solar Cells. Joule 3, 387–401 (2019) to estimate the mismatch of the current limiting junction.

Response:

In this paper, McMeekin et al. calculated the mismatch factor for the current-limiting junction of tandem/triple cells under the xenon arc lamp spectrum. The authors then adjusted the lamp intensity to the initial mismatch factor and performed the $J-V$ measurement. However, unless the mismatch factors for all the sub-cells are equal with respect to such a single-source sun simulator, one of the sub-cells would be inevitably illuminated under a deviated spectrum compared to AM1.5G, which could influence the shape of the $J-V$ curve and result in an over-

/underestimated FF and hence the PCE (Timmreck et al. *Nat. Photon.* 9, 478-479 (2015), see discussions in Supporting Information, section 3.2). As is also discussed in McMeekin’s paper, although the spectral difference between the lamp and AM1.5G is calculated, the different current matching conditions under the different spectra make it impossible to determine the FFs of tandem/triple devices correctly. Instead, the authors simply corrected the J_{sc} and PCE by dividing the mismatch factors measured for both the tandem (1.004) and triple cell (1.486).

In our approach, we accurately determined the current-limiting sub-cell and the corresponding J_{sc} by measuring the spectral response (and EQE) of all the sub-cells under different biasing conditions (Following the method reported by Timmreck et al. *Nat. Photon.* 9, 478-479 (2015)). We then adjusted the lamp intensity of our solar simulator such that the $J-V$ measurement delivers a J_{sc} equivalent to that of the current-limiting sub-cell measured from EQE. To see whether the spectral deviation of our solar simulator (TUE) has a significant impact on the shape of the $J-V$ curve for a triple cell, we performed $J-V$ analysis on the same triple-junction cell using a more reliable solar simulator (equipped with individual LEDs, with spectral mismatch $\sim 1\%$ for the current limiting 1.23 eV sub-cell). As can be seen from the figure below, both measurements show very similar $J-V$ characteristics, with a comparable V_{oc} and FF, suggesting that the over-/underestimation of FF should be relatively small in our case.

Cell	$J_{AM1.5G}$ (mA cm ⁻²)	$J_{Simulator}$ (mA cm ⁻²)	Mismatch
Si ref	32.44	32.83	
1.73 eV	8.65	8.41	0.961
1.57 eV	9.18	9.42	1.014
1.23 eV	7.92	7.90	0.986

9. Clarification is needed on whether solutions were weighed and stirred in the glovebox. Giving precise details such as these really help other researchers in the field to follow the work.

Response:

Following the reviewer’s suggestion, we added more details of solution preparations in Method section (Page 16):

We note that all materials except PEDOT:PSS and PTAA (stored in ambient) were stored in a dry N₂ glovebox. While it was crucial to weigh Sn-containing compounds in a dry N₂ glovebox, other compounds showed little influence on the device performance when weighed either in dry N₂ or in an ambient atmosphere. All solvents for spin coating were stored in an N₂ glovebox. All solutions were prepared and stirred in the same glovebox and were cooled to room temperature before use.

10. “A triple device comprising a 1.73 eV, 1.57 eV, and 1.23 eV sub-cells shows a promising

PCE of 16.8%, with very low potential energy drop ($V_{oc} = 2.78$ V) and resistivity loss ($FF = 0.81$)” It is suggested that the actual potential drop (70 mV) be mentioned; something like: “A triple device comprising a 1.73 eV, 1.57 eV, and 1.23 eV sub-cells shows a promising PCE of 16.8%, with very low potential energy drop of only 70mV in comparison to the sum of the V_{oc} of each sub cell ($V_{oc} = 2.78$ V) and low resistivity loss ($FF = 0.81$)”

Response:

We rephrased the sentence in the Conclusion (Page 15) as follows:

A triple device comprising a 1.73 eV, 1.57 eV, and 1.23 eV sub-cells shows a promising PCE of 16.8%, with very low potential energy drop of only 80 mV in comparison to the summed V_{oc} of all three sub-cells ($V_{oc} = 2.78$ V) and low resistivity loss ($FF = 0.81$).

11. Simulation: Mention all the layer thicknesses used, and report the n,k values where measured. If taken from literature, provide the appropriate references.

Response:

Following the reviewer’s suggestion, we have plotted the n, k values of all layers used for simulation in Supplementary Fig. 20 on Page S14. The corresponding layer thicknesses used for optical simulation have been added to the legend of Supplementary Fig. 14 and 19.

Reviewer #2 (Remarks to the Author):

The processing of lead halide perovskites using two-step sequential method has been widely reported and investigated in the community. I do not see much new ideas herein. The processing of mixed Sn-Pb narrow-bandgap perovskite has been, however, rarely reported. Compared to authors’ recent work on two-step processed Sn-Pb perovskite solar cells (DOI: 10.1002/aenm.202000566), I did not find any advances in either performance or processing approach itself. The design of interlayer using ALD deposited SnO₂ and evaporated Au has also been reported in previous work (10.1038/s41560-019-0466-3). For the tandem cells herein, the demonstrated PCEs in this work are much lower than those of previous works (e.g.10.1038/s41560-019-0466-3; 10.1126/science.aav7911 and others). I would be happy to reconsider this work, if the authors could demonstrate PCEs of Sn-Pb single junction cell and tandem cells comparable or even higher than previous works.

Response:

We acknowledge that the two-step spin coating technique that we used in this study is not new to the research community and that our recorded PCEs of single-junction and tandem solar cells are lower than previous results in the literature.

While the figure of merit for a solar cell is its PCE, for the first time, we have presented a straightforward pathway to developing monolithic all-perovskite triple-junction solar cells at a relevant performance level (16.8%). A two-step deposition was used to fabricate three different perovskite absorbers (1.73/1.57/1.23 eV) with similar processing protocols, which is highly desired considering the complexity of the multijunction device stack. Meanwhile, a detailed optimization and electrical/optical loss-analysis of the recombination layers was presented, which is usually not discussed in detail in other tandem studies.

In all, despite the efficiency of the single-junction and tandem solar cells presented in this study, we believe that our method shows a promising approach towards high-efficiency all-perovskite multijunction solar cells and is of interest to the research community.

Other comments:

The author chose FA/MA as the main composition for the wide-bandgap and mid-bandgap perovskites. Would following processing affect the performance of MA-containing materials? It seems that MA-incorporated perovskites are not very stable under thermal stressing. Another issue is the photo-induced instability of MA-contained wide-bandgap perovskites.

Response:

Indeed, it is widely accepted in the research society that MA-based perovskites are less stable concerning humidity and thermal stress. However, the use of MA broadens the flexibility for composition engineering, and most of the top-performing PSCs to date are still based on MA-containing mixed-cation recipes. A recent study found that MA-containing PSCs could also pass all the damp heat, thermal cycling and humidity freeze tests for commercial modules, provided that an effective encapsulation is applied on the top to suppress outgassing of volatile decomposition products from the perovskites (Shi et al. Science, 2020, 368, eaba2412).

Similarly, our SALD-SnO₂ layer should also improve the ambient and thermal stability of the perovskite layer for its excellent permeation barrier property. To study the impact of thermal stressing on the performance of perovskite layers, we fabricated single-junction PSCs based on a device structure of ITO/PTAA/perovskite (1.73 eV or 1.57 eV)/C₆₀/SALD-SnO₂/Al. After depositing the SALD-SnO₂, the samples were subjected to the same annealing procedures used for PEDOT:PSS (air) and perovskite layers (N₂) during a multijunction cell fabrication: For the 1.73 eV cell, the substrates were annealed at 100 °C either for 60 min. in N₂ or 15 min. in air + 30 min. in the N₂ + 15 min. in air + 30 min. in N₂; For the 1.57 eV cell, the substrates were annealed at 100 °C either for 30 min. in N₂ or 15 min. in air + 30 min. in N₂. The devices were then finalized by evaporating Al contact on the top. From the $J-V$ characteristics (now added in Supplementary Fig. 17), all the thermally treated 1.73 eV- and 1.57 eV-PSCs exhibited almost identical PV performance compared to their reference cells prepared in the same batch. Furthermore, XRD measurements suggest that no distinct degradation (formation of PbI₂) was found after thermal stressing. The result suggests that the followed (annealing) processes of multiple layers should not affect the performance of the bottom MA-containing perovskites.

As for the photo-stability of the wide-bandgap perovskite, we carried out PL measurements of a Cs_{0.1}(FA_{0.66}MA_{0.34})_{0.9}PbI₂Br thin-film during 120 min of light exposure (405 nm LED). No redshift of the emission peak was found during light soaking, suggesting that the film was stable during the tracking period. This information has now been added to Supplementary Fig. 2.

The bandgap (~1.73 eV) of wide-bandgap subcells used for the triple-junction is far from ideal situation (~2.0 eV). Authors should consider fabricating triple-junction cells with better bandgap matching. Otherwise, it loses the benefits of making triple-junction configuration.

Response:

Indeed, as discussed in the manuscript, an efficient ~2.0 eV wide-bandgap perovskite sub-cell is ultimately needed to fabricate a realistic triple-junction solar cell (with PCEs higher than both the single-junction and tandems). However, the performance of such wide-bandgap cells to date is suboptimal. Our survey of the literature (now added in Supplementary Fig. 21 and Supplementary Table 5) suggests that the V_{oc} deficit ($E_g/q - V_{oc}$) is significantly increased (> 600 mV) for perovskites with bandgaps of above 1.8 eV. Such a high loss-in-potential likely originates from trap-assisted charge recombination in the perovskite bulk and energy misalignment between the perovskite and charge transport layers. More importantly, the photo-stability of Br-rich systems is inferior to that of the I-rich perovskite materials.

In a series-connected multijunction solar cell, the voltage is added up from all the sub-cells at equal currents. Therefore, using a 2.0 eV wide-bandgap perovskite with a large V_{oc} -loss and low FF will not necessarily improve the performance of the current triple-junction solar cell.

In Supplementary 2, we have added the PV characteristics of a 2.0 eV wide-bandgap (~380 nm) single-junction cell (with Al top contact) prepared by ourselves. This cell has a PCE of 5.6%, with a J_{sc} of 8.3 mA cm⁻² (corrected by EQE). Compared to the 1.73 eV perovskite, the 2.0 eV cell is less stable and shows a more pronounced hysteresis effect. The V_{oc} of 1.11 V (after 300 s stabilization) is comparable to the 1.73 eV sub-cell, but the FF is much lower (0.61), which makes it less desirable in the current triple-junction cell.

Nevertheless, we would like to point out that despite the efficiency of the current design is limited by the spectral overlap, our approach is still valid and highlights the importance of ICLs in developing highly efficient multijunction cells. With further development of wide bandgap perovskite materials with a small V_{oc} deficit and good photo-stability, high-efficiency all-perovskite triple-junction solar cells should be easily accessible.

Reviewer #3 (Remarks to the Author):

[1] On Page 9, the authors attribute the reduced J_{sc} (from 14 to 13.1 mA/cm²) in the 1.23-eV subcell of the C60-based tandem devices is attributed to the change in optical interference. Do the changes in the EQE interference agree with the simulation?

Response:

We have now added the simulated EQE data to Supplementary Fig. 12 on Page S8.

The optical simulation also confirms that the shift of the EQE spectra (1.23 eV sub-cell) corresponds to the change from PCBM to C₆₀ in the ICLs (Fig. 2a and 2c). This originates from the change in optical interference. As a result, the photocurrent in the 1.23 eV sub-cell is reduced from 14 to 13 mA cm⁻².

[2] On Page 13, PTAA was deposited on PEDOT:PSS in the 1.57 eV mid subcell. Did the authors test the direct replacement of PEDOT:PSS by PTAA for the mid subcell?

Response:

Follow the reviewer's advice, we included the 1.57 eV single-junction solar cells with different bottom contacts: (a) ITO/PEDOT:PSS (and/or PTAA), (b) ITO/SALD-SnO₂/PEDOT:PSS (and/or PTAA), and (c) ITO/SALD-SnO₂/Au/PEDOT:PSS (and/or PTAA). In this way, we could already check the effectiveness of the interconnecting layers for a potential multijunction stack.

As expected, in configuration a, the PTAA layer alone outperforms PEDOT:PSS due to its better hole selectivity towards 1.57 eV perovskite material. When a PEDOT:PSS/PTAA stack was used in the device, only the J_{sc} was slightly reduced as a result of parasitic absorption and reflection from PEDOT:PSS.

In configuration b and c, again, the SALD-SnO₂/(Au)/PEDOT:PSS layer stack displays low performance due to the poor hole-selectivity of PEDOT:PSS. However, suboptimal device performance is also seen when using SALD-SnO₂/(Au)/PTAA as the bottom contact. This can be attributed to the low conductivity of our (undoped) PTAA hole transport layer, which creates an electronic barrier at the SnO₂/(Au)/PTAA interface. In the end, we found that a combined SALD-SnO₂/(Au)/PEDOT:PSS/PTAA works the best in such devices.

We have now added this information to Supplementary Fig. 16 on Page S11.

REVIEWERS' COMMENTS

Reviewer #1 (Remarks to the Author):

The main novelty of this paper lies in the demonstration of a triple-junction all perovskite device of reasonable efficiency (16.8%). This is a significant improvement upon the only other available demonstration of 6.7%. Their realisation of a high quality interlayer which nearly eliminates electronic loss is also very valuable. The paper does not skimp on detail, making it convenient for other researchers to reproduce. In short, this is a fantastic paper which in my view should be published in Nature Communications. There are few minor issues which should be addressed before publication:

1. "Previously, it has been demonstrated that a SnO₂ layer grown by atomic layer deposition (ALD) can prevent solvent damage without using the sputtered indium tin oxide (ITO)." It would be constructive to compare the recombination layers in the literature, explaining the different sputtered layers which have been used, to the recent ALD SnO₂ development, with and without ITO. I am not asking for more experiments, just a bit of an expansion on the discussion of this point. It is very interesting that SnO₂ + very thin (non continuous) Au works, but optically what is best?
2. Explain advantages of S-ALD compared to conventional thermal ALD/PEALD eg. Reduction of ALD cycle time, potential processing at lower temperatures. Refer to use of S-ALD in literature for perovskite devices. Examples below:
 - L. Hoffmann et al., "Spatial Atmospheric Pressure Atomic Layer Deposition of Tin Oxide as an Impermeable Electron Extraction Layer for Perovskite Solar Cells with Enhanced Thermal Stability", ACS Appl. Mater. Int., 10, 6006-6013, 2018
 - M. Najafi et al., "Highly Efficient and Stable Semi-Transparent p-i-n Planar Perovskite Solar Cells by Atmospheric Pressure Spatial Atomic Layer Deposited ZnO," Sol. RRL, vol. 1800147, 1800147, 2018
 - V. Zardetto et al., "Towards Large Area Stable Perovskite Solar Cells and Modules," IEEE PVSC, 978-1-7281-0494-2, 2019
3. Clearly state the cell aperture area (0.09 cm² versus 0.16 cm²) for the PCE achieved for single junction, tandem and triple junction devices. Also provide your cell layout with dimensions in the supplementary. This would be important for researchers trying to reproduce your high fill factors.
4. Include J-V curves in the supplementary information showing the increased device performance achieved with PTAA deposited on top of PEDOT:PSS for the 1.57 eV perovskite absorber device versus PTAA.
5. It would be useful to know the reasoning for the DMF:DMSO volume ratio of ~10.1:1?
6. In experimental methods section it is important to provide the growth parameters and details for S-ALD SnO₂:
 - Deposition temperature
 - Exposure time of precursors
 - GPC, number of ALD cycles per rotation of substrate, substrate velocity/rotation frequency
 - What carrier gas was utilised and it's flow rate
 - At what temperature were precursor vessels held, was a bubbler utilized and the precursor partial pressures
7. Please mention the sheet resistance of the ITO used.
8. Spectral mismatch corrected PCE has to be reported. Spectral mismatch can cause significant

overestimation of PCE in multi-junctions. The authors could follow the approach shown by McMeekin, D. P. et al. Solution-Processed All-Perovskite Multi-junction Solar Cells. *Joule* 3, 387–401 (2019) to estimate the mismatch of the current limiting junction.

9. Clarification is needed on whether solutions were weighed and stirred in the glovebox. Giving precise details such as these really help other researchers in the field to follow the work.

10. "A triple device comprising a 1.73 eV, 1.57 eV, and 1.23 eV sub-cells shows a promising PCE of 16.8%, with very low potential energy drop ($V_{oc} = 2.78$ V) and resistivity loss ($FF = 0.81$)" It is suggested that the actual potential drop (70 mV) be mentioned; something like: "A triple device comprising a 1.73 eV, 1.57 eV, and 1.23 eV sub-cells shows a promising PCE of 16.8%, with very low potential energy drop of only 70mV in comparison to the sum of the V_{oc} of each sub cell ($V_{oc} = 2.78$ V) and low resistivity loss ($FF = 0.81$)"

11. Simulation: Mention all the layer thicknesses used, and report the n,k values where measured. If taken from literature, provide the appropriate references.

Reviewer #3 (Remarks to the Author):

The authors have very well addressed my comments. I do not have further comments/suggestions to make.

We again thank the reviewer for his/her valuable comment and effort to improve our manuscript. Below we repeat the and concern of the reviewer (in red font) and include our response and outline the changes made in the revised manuscript (highlighted).

Reviewer #1 (Remarks to the Author):

In response 5, they state:

"In this study, we use $\sim 1.25 \text{ mol L}^{-1}$ of PbI_2 (or $\text{Pb}_{0.5}\text{Sn}_{0.5}\text{I}_2$) precursor solution in the first step. For 553 mg PbI_2 , 1:1 molar ratio of PbI_2 :DMSO was formed by adding $\sim 86 \text{ mL}$ DMSO, then $\sim 876 \text{ mL}$ DMF was used to develop a total solution concentration of $\sim 1.25 \text{ M}$.

Therefore, the volume ratio of DMF:DMSO is close to 10.1:1. To clarify this, we added a note in the Methods section"

The note they added does not contain all the information required, they should add this detailed description of the way they have added the DMSO to the PbI_2 into the methods section, the order of solvent addition may make a difference, and they should be as transparent as possible.

Response:

Thank you for pointing this out. We further clarify our procedures in the Methods section, on Page 15:

For 1.73 eV $\text{Cs}_{0.1}(\text{FA}_{0.66}\text{MA}_{0.34})_{0.9}\text{PbI}_2\text{Br}$, 876 μL DMF (99.8%) and 86.4 μL DMSO (99.9%) were sequentially added to 34.8 mg CsI (Sigma-Aldrich, 99.999%) and 553 mg PbI_2 (Sigma-Aldrich, 99.999%) to form a $\sim 1.25 \text{ M}$ precursor solution, before stirring at $60 \text{ }^\circ\text{C}$ overnight. The small portion of DMSO was added to form a 1:1 molar ratio of PbI_2 :DMSO. We note that the order of solvent addition (DMF and DMSO) does not change the device performance.